# ATTACKING BINARIZED NEURAL NETWORKS

**Angus Galloway[1], Graham W. Taylor[1,2,3] Medhat Moussa[1]**
[1]School of Engineering, University of Guelph, Canada
[2]Canadian Institute for Advanced Research
[3]Vector Institute for Artificial Intelligence, Canada
`{gallowaa,gwtaylor,mmoussa}@uoguelph.ca`

## ABSTRACT

Neural networks with low-precision weights and activations offer compelling efficiency advantages over their full-precision equivalents. The two most frequently discussed benefits of quantization are reduced memory consumption, and a faster forward pass when implemented with efficient bitwise operations. We propose a third benefit of very low-precision neural networks: improved robustness against some adversarial attacks, and in the worst case, performance that is on par with full-precision models. We focus on the very low-precision case where weights and activations are both quantized to $\pm 1$, and note that stochastically quantizing weights in just one layer can sharply reduce the impact of iterative attacks. We observe that non-scaled binary neural networks exhibit a similar effect to the original *defensive distillation* procedure that led to *gradient masking*, and a false notion of security. We address this by conducting both black-box and white-box experiments with binary models that do not artificially mask gradients.[1]

## 1 INTRODUCTION

The ability to fool machine learning models by making small changes to their input severely limits their potential for safe use in many real-world scenarios. Example vulnerabilities include a seemingly innocuous audio broadcast that is interpreted by a speech recognition model in a smartphone, with the intent to trigger an e-transfer, as well as pictures or identity documents that are automatically tagged as someone other than the real individual.

The two most common threat models when evaluating the security of a system are the *black-box* and *white-box* assumptions, which represent varying degrees of information that an adversary may possess. In a *black-box* threat model, an adversary has similar abilities to a normal user that interacts with a system by providing inputs and observing the corresponding outputs. Under this threat model, an adversary generally does not know details of the model architecture or dataset used to train the model. Of course, an adversary is free to assume that a convolutional architecture was likely used if the input domain is images, or a recurrent model for speech or text.

In a *white-box* threat model, an adversary has complete access to the model architecture and parameters. In the case of neural networks, white-box attacks frequently rely on gradient information to craft especially strong *adversarial examples*, where *strong* means that the example is very close to the original input as defined by some distance norm (e.g. $L_0$–number of features modified, $L_2$–mean squared distance), yet is very likely to cause the model to yield the incorrect output. For both threat types, targeted attacks where a model is made to fail in a specific way (e.g. causing a handwritten '7' look like a '3') represents a stronger attack than simple misclassification.

The problem with deploying machine learning systems that are *secured* in a traditional sense, is that adversarial examples have been shown to generalize well between models with different source and target architectures (Szegedy et al., 2013; Papernot et al., 2017b; Tramèr et al., 2017). This means that a secured model can be compromised in an approximately white-box setting by training and attacking a substitute model that approximates the decision boundary of the model under attack (Papernot et al., 2017b). Thus, to make strong conclusions about the robustness of a machine learning model to adversarial attacks, both threat models should be considered.

---

[1]Source code available at `https://github.com/AngusG/cleverhans-attacking-bnns`

Tangent to research on defences against adversarial attacks, significant progress has been made towards training very low-precision deep neural networks to accuracy levels that are competitive with full-precision models (Courbariaux & Bengio, 2016; Zhou et al., 2016; Tang et al., 2017). The current motivation for extreme quantization is the ability to deploy these models under hardware resource constraints, acceleration, or reduced power consumption. Ideally, $32\times$ compression is possible by using 1-bit to represent single-precision floating point parameters. By similarly quantizing activations, we can reduce run-time memory consumption as well. These savings enable large scale deployment of neural networks on the billions of existing embedded devices. Very low-precision models were designed with deployment in mind, and may be responsible for making critical decisions in embedded systems, all subject to reverse engineering and a diverse set of real world attacks. With much at stake in applications like autonomous navigation, robotics, and network infrastructure, understanding how very low-precision neural networks behave in adversarial settings is essential. To that end, we make the following contributions:

- To the best of our knowledge, we are the first to formally evaluate and interpret the robustness of binary neural networks (BNNs) to adversarial attacks on the MNIST (LeCun & Cortes, 1998) and CIFAR-10 (Krizhevsky, 2009) datasets.

- We compare and contrast the properties of low-precision neural networks that confer adversarial robustness to previously proposed defense strategies. We then combine these properties to propose an optimal defense strategy.

- We attempt to generalize and make recommendations regarding the suitability of low-precision neural networks against various classes of attacks (e.g. single step vs. iterative).

## 2 BACKGROUND

Since the initial disclosure of adversarial examples by Szegedy et al. (2013) and Biggio et al. (2013), many defense strategies have been proposed and subsequently defeated. It is generally accepted that strategies for mitigating the impact of these examples still lag behind state of the art attacks, which are capable of producing adversarial examples that are indistinguishable from unmodified inputs as perceived by humans. In general, there are two approaches to defending against adversarial examples: *reactive*–detecting the presence of adversarial examples, such as through some notion of confidence-based outlier detection. On the other hand, a *proactive* approach aims to improve the robustness of the underlying model, which may involve adding an extra class to which malicious inputs should be assigned (Papernot & McDaniel, 2017). The latter approach is important for building reliable systems where a sensible decision *must* be made at all times. In this work, we focus solely on the proactive approach.

To define adversarial examples, we require some measurement of distance that can be computed between perturbed inputs and naturally occurring inputs. In the visual domain, it is convenient if the metric approximates human perceptual similarity, but is not required. Various $L_p$ norms have been used in the literature: $L_0$–number of features modified, $L_2$–mean squared distance, $L_\infty$–limited only in the maximum perturbation applied to any feature. We evaluate at least one attack that is cast in terms of each respective distance metric, and leave discussion of the optimal distance metric to future work.

The most compelling explanation for the existence of adversarial examples proposed to date is the linearity hypothesis (Goodfellow et al., 2015). The elementary operators, matrix dot-products and convolutions used at each layer of a neural network are fundamentally too linear. Furthermore, the non-linearity applied at each layer is usually itself either piecewise linear (e.g. ReLU), or we have specifically encouraged the network through initialization or regularization to have small weights and activations such that its units (e.g. sigmoid, tanh) operate in their linear regions. By adding noise to inputs which is highly correlated with the sign of the model parameters, a large swing in activation can be induced. Additionally, the magnitude by which this noise must be scaled to have this effect tends to diminish as the input dimensionality grows. This piecewise linearity also makes neural networks easy to attack using the gradient of the output with respect to the input, and consequently, the resulting incorrect predictions are made with high-confidence.

Fortunately, we are reminded that the universal approximation theorem suggests that given sufficient capacity, a neural network should at least be able to represent the type of function that resists

adversarial examples (Goodfellow et al., 2015). The most successful defense mechanism to date, adversarial training, is based on this premise, and attempts to learn such a function. The *fast gradient sign method* (FGSM) is one such procedure for crafting this damaging noise, and is still used today despite not being state-of-the-art in the white-box setting, as it is straightforward to compute, and yields examples that transfer well between models (Goodfellow et al., 2015).

The linearity hypothesis was one of the main reasons for initially considering binarized neural networks as a natural defense against adversarial examples. Not only are they highly regularized by default through severely quantized weights, but they appear to be more non-linear and discontinuous than conventional deep neural networks (DNNs). Additionally, we suspect that the same characteristics making them challenging to train, make them difficult to attack with an iterative procedure. At the same time, assumptions regarding the information required by an effective adversary have become more and more relaxed, to the extent that black-box attacks can be especially damaging with just a small set of labeled input-output pairs (Papernot et al., 2017b).

Perhaps the most striking feature of adversarial examples is how well they generalize between models with different architectures while trained on different datasets (Goodfellow et al., 2015; Papernot et al., 2017b; Kurakin et al., 2017a). It was shown by Kurakin et al. (2017b) that 2/3 of adversarial ImageNet examples survive various camera and perspective transformations after being printed on paper and subsequently photographed and classified by a mobile phone.

The most successful black-box attacks have the secured model (Oracle) assign labels to a set of real or synthetic inputs, which can be used to train a substitute model that mimics the Oracle's decision boundary (Papernot et al., 2017b). A single step attack, such as FGSM, can be used on the smooth substitute model to generate examples that transfer, without having access to the original training data, architecture, or training procedure used by the Oracle. Papernot et al. (2017b) showed they are able to compromise machine learning models 80% of the time on small datasets like MNIST using various shallow MLP-based substitute models. There is not a particularly high correlation between test accuracy and transferability of adversarial examples; therefore despite not attaining great results on the original MNIST task, a simple substitute learns enough to compromise the Oracle. This technique was shown to overcome gradient masking approaches, such as in the case with models that either obscure or have no gradient information, such as k-nearest neighbors or decision trees.

With strong adversarial training of the model to be defended, attacks generated using the substitute model do not transfer as well. Therefore, to be compelling, BNNs should be able to handle training with large $\epsilon$ while maintaining competitive test accuracy on clean inputs relative to full-precision.

The strongest white-box attacks all use an iterative procedure; however, the resulting examples do not transfer as well as single step methods (Goodfellow et al., 2015). An iterative attack using the Adam optimizer was proposed by Carlini & Wagner (2017) that outperforms other expensive optimization based approaches Szegedy et al. (2013), the Jacobian-based saliency map attack (JSMA) (Papernot et al., 2015), and Deepfool (Moosavi-Dezfooli et al., 2015) in terms of three $L_p$ norms previously used as an adversarial example distance metrics in the literature. We have made our best attempt to use state-of-the-art attacks in our experiments.

## 3 EXPERIMENTS

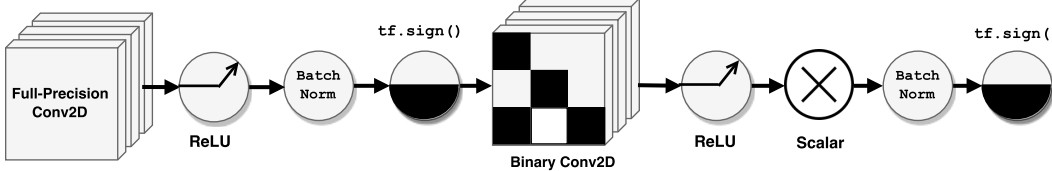

Figure 1: Blocks used in binary convolution architecture.

In Figure 1, we depict the quantization scheme applied to the base convolutional neural network provided in the CleverHans library tutorials (Papernot et al., 2017a). In the first layer, we retain weights and activations in single-precision floating point. Weights in hidden layers are binarized either de-

terministically or stochastically, as in Courbariaux & Bengio (2016), and activations were always binarized deterministically. Unlike in Courbariaux & Bengio (2016), we stochastically quantize weights at *test* time as a possible defense against iterative attacks. Under the stochastic binarization scheme, weights are sampled once per forward pass from a Bernoulli distribution with probability given by passing the real valued weight through the hard sigmoid function from Courbariaux & Bengio (2016). Lastly, we map the Bernoulli samples $\in [0, 1]$ to $\pm 1$ by multiplying by 2 and subtracting $1^2$.We do not find that this significantly slows down training with TensorFlow (Abadi et al., 2015) on a modern GPU, but these networks take between 3–4$\times$ as many epochs as a deterministically quantized binary network to converge.

We use the straight through estimator (STE) to back-propagate gradients through the quantization step (Bengio et al., 2013). We optionally insert a small (e.g 1e-2) tunable scalar after the ReLU in hidden layers, to compensate for an increase in the L1 norm of the activations due to binarization. Tang et al. (2017) also used this approach to reach similar accuracy gains as those conferred by the more expensive XNOR-Net channel-wise normalization scheme (Rastegari et al., 2016). Convolution kernels were initialized from a truncated normal distribution with $\sigma$=0.2 for accumulating full-precision weight updates, and were quantized to $\pm 1$ in the forward pass. Batch normalization was applied before quantizing activations to ensure they were centered around zero (Ioffe & Szegedy, 2015).

We report test error rates for these models on MNIST (LeCun & Cortes, 1998) with varying capacity in Table 6 of Appendix A. Capacity is denoted by the number of kernels in the first layer, $K_{Layer1}$. All subsequent layers had exactly double this number of kernels. Models were trained for 15 epochs unless indicated otherwise. In general, models with full-precision weights and activations under-fit the naturally occurring data less than a binary equivalent, with error rates of approximately 1% and 2%, respectively. With the addition of the small learned scaling factor, the binary models converge to approximately the same error rate as the full-precision model on MNIST and CIFAR-10.

We experiment with three different types of adversarial training, depending on the combination of dataset and attack: FGSM with fixed $\epsilon$, FGSM with $\epsilon$ sampled from a truncated normal distribution as in Kurakin et al. (2017a), and projected gradient descent (PGD) (Madry et al., 2017), which is the state-of-the-art adversarial training procedure for MNIST. We do not necessarily pair all training methods against all attacks. The model's own best prediction is used as the true label to minimize in adversarial training, unless otherwise noted to prevent the *label leaking* effect (Kurakin et al., 2017a). We first attempt to fool our binarized networks with single step attacks in a white-box setting, and progressively scale up to stronger state-of-the-art attacks. All experiments were conducted by seeding the TensorFlow random number generator with the value 1234.

## 3.1 WHITE-BOX ATTACKS

All experiments were conducted in TensorFlow, and used either v2.0.0 of CleverHans (Papernot et al., 2017a), or Foolbox, a Python toolbox for creating adversarial examples (Rauber et al., 2017). All attacks were clipped to the anticipated input range during adversarial training and evaluation. For single step attacks, we fix the magnitude of the perturbation and attack the whole test set, then report accuracy on the new test set. The general procedure for iterative attacks is to fix the step size per iteration or learning rate, and number of iterations. We then report accuracy on the perturbed test set after this many iterations while keeping other hyper-parameters constant.

### 3.1.1 FAST GRADIENT SIGN METHOD

The FGSM is a simple but effective single step attack first introduced in Goodfellow et al. (2015), and defined in eq (1). The attack linearly approximates the gradient of the loss used to train the model with respect to the input. The gradient is thresholded by taking its sign, scaled by a uniform constant $\epsilon$, and added to, or subtracted from, the input, depending on if we wish to minimize the current class, or move in the direction of a target class:

$$x_{adv} = x + \epsilon \times sign(\Delta_x J(\theta, x, y)) \tag{1}$$

---

[2]In TensorFlow this can be accomplished with:
```
2 * Bernoulli(probs=tf.clip_by_value((x + 1.)/ 2., 0., 1.))).sample() -1
```

To confer robustness to more than one value of $\epsilon$ with which an adversary may use to attack, the adversarial training procedure from Kurakin et al. (2017a) proposes to sample a unique $\epsilon$ for each training example from a truncated normal distribution. We set the standard deviation to $\sigma = ceil(\epsilon_{\max} * 255/2)$. We consider up to $\epsilon_{\max} = 0.3$, as this is a common upper limit for a $L_\infty$ norm perturbation that is not easily perceived by humans, and corresponds to a 30% change in pixel intensity for an arbitrary number of pixels.

Table 1: Accuracy on adversarial examples generated with a FGSM misclassification attack on the MNIST test set with three values of $\epsilon$. Three different models were evaluated: A is full-precision, B is binary, and C is binary with a learned scalar. Models trained with, and without, adversarial training are shown. The '+' suffix indicates the model was trained for the last 5 epochs with the procedure from Kurakin et al. (2017a). All values averaged over four runs for models trained from scratch.

| Model | $K_{Layer1}$ | $\epsilon = 0.1$ | $\epsilon = 0.2$ | $\epsilon = 0.3$ |
|---|---|---|---|---|
| | 64 | 74±4% | 39±4% | 22±5% |
| A | 128 | 75±3% | 34±2% | 18±3% |
| | 256 | 74±1% | 33±2% | 17±3% |
| | 64 | 75±2% | 64±3% | 59±2% |
| B | 128 | 85±1% | 77±2% | 70±2% |
| | 256 | **89±1%** | **83±1%** | **78±1%** |
| | 64 | 56±7% | 27±5% | 15±3% |
| C | 128 | 64±3% | 26±9% | 11±5% |
| | 256 | 73±2% | 37±6% | 16±3% |
| | 64 | 80±1% | 62±1% | 63±1% |
| A+ | 128 | **83±1%** | **71±1%** | **72±1%** |
| | 256 | **83±1%** | **71±2%** | 70±2% |
| | 64 | 68±1% | 32±5% | 31±5% |
| B+ | 128 | 75±1% | 50±3% | 45±4% |
| | 256 | 79±2% | 64±3% | 58±2% |
| | 64 | 80±2% | 47±7% | 38±4% |
| C+ | 128 | 82±1% | 50±3% | 40±2% |
| | 256 | **84±3%** | 54±4% | 41±4% |

In Table 1, it can be observed that a plain binary network without adversarial training (B) achieves the best robustness to FGSM, with nearly 90% accuracy for $\epsilon = 0.1$ for the highest capacity model. We postpone a formal explanation of this outlier for the discussion. Our results for large $\epsilon$ agree with observations made by Madry et al. (2017) where they found FGSM to be suboptimal for training as it yields a limited set of adversarial examples. We suspect that the reason neither scaled nor unscaled binary models performed well when trained with an adversary and tested on larger values of $\epsilon$ is because by the time adversarial training was introduced at epoch 10, both had entered into a state of decreased learning. Our binary weight implementation makes updates to real valued weights during training, which are binarized in the forward pass. The real valued weights tend to polarize as the model converges, resulting in fewer sign changes. Regularization schemes actually encouraging the underlying real valued weights to polarize around ±1 have been proposed (Tang et al., 2017), but we do not find this to be particularly helpful after sweeping a range of settings for the regularization constant $\lambda$. Regardless, in this case, the binary models did not benefit from adversarial training to the same extent that the full-precision models did.

We find that adversarial training with binary models is somewhat of a balancing act. If a strong adversary is introduced to the model too early, it may fail to converge for natural inputs. If introduced too late, it may be difficult to bring the model back into its *malleable* state, where it is willing to flip the sign of its weights. Despite this challenge, the *scaled* binary model (C+) (see Figure 1 for location of optional scalar) reaped significant benefits from adversarial training and its accuracy was on par with the full-precision model for $\epsilon = 0.1$.

To investigate the low performance observed against large $\epsilon$ in Table 1, models A and C were trained from scratch with 40 iterations of PGD (Madry et al., 2017). Table 2 shows the result of this new training and subsequent FGSM attack performed identically to that of Table 1. A similar trend was found in Tables 1 and 2, where the lowest capacity models struggle to become robust against large

$\epsilon$. Once the scaled binary model had sufficient capacity, it actually slightly outperforms its full-precision equivalent for all values of $\epsilon$. With this, we have demonstrated that not only can BNNs achieve competitive accuracy on clean inputs with significantly fewer resources, but they can also allocate excess capacity in response to state-of-the-art adversaries.

Table 2: Accuracy on adversarial examples generated with a FGSM misclassification attack on the MNIST test set with three values of $\epsilon$. Both full-precision (A+*) and scaled binary (C+*) models were trained with 40 iterations of PGD (Madry et al., 2017) for the last 5 epochs with with $\epsilon = 0.3$. All values averaged over four runs for models trained from scratch.

| Model | $K_{Layer1}$ | $\epsilon = 0.1$ | $\epsilon = 0.2$ | $\epsilon = 0.3$ |
|-------|------|------|------|------|
|       | 64   | 94.7±0.2% | 90.9±0.3% | 80.2±0.2% |
| A+*   | 128  | 95.8±0.3% | 92.3±0.3% | 82.9±0.9% |
|       | 256  | 95.9±0.2% | 92.9±0.3% | 85±1% |
|       | 64   | 92.9±0.4% | 83.6±0.6% | 67±2% |
| C+*   | 128  | 95.0±0.2% | 88.2±0.3% | 74.3±0.6% |
|       | 256  | **96.8±0.3%** | **93.4±0.3%** | **85.6±0.6%** |

### 3.1.2 CARLINI-WAGNER ATTACK CARLINI & WAGNER (2017)

The Carlini-Wagner L2 attack Carlini & Wagner (2017) (CWL2) is an iterative process guided by an optimizer such as Adam, that produces strong adversarial examples by simultaneously minimizing distortion, and manipulating the logits per the attack goal. We use the implementation from CleverHans (Papernot et al., 2017a) and show results in Table 3 and Figure 2. Only binary models are shown in Table 3 because all but two full-precision models had *zero* accuracy after running CWL2 for 100 iterations. The best full-precision model was A256+ with 1.8±0.9% accuracy. We note that the stochastically quantized binary models with scaling to prevent gradient masking ('S' prefix) underfit somewhat on the training set, and had test error rates of 8±1%, 5±2%, and 3±1% for each of S64, S128, and S256 averaged over four runs. For S256, this test error can be compared with an unscaled binary model which only achieves 22±3% accuracy *with* gradient masking compared to 46±3% *without*.

Table 3: Carlini-Wagner $L_2$ targeted attack on MNIST test set (90k images total) for binary models versus increasing capacity from left to right. All attacks were run for 100 iterations as all full-precision models were driven to have zero accuracy by this point. Models with 'S' prefix used stochastic quantization.

| Model | B32 | B64 | B128 | B256 |
|-------|------|------|------|------|
| Accuracy | **7±1%** | 7±3% | 12±3% | 22±3% |
| Mean $L_2$ dist. | 2.88±0.02 | 3.1±0.2 | 3.2±0.1 | 3.2±0.1 |
| Model | B32+ | B64+ | B128+ | B256+ |
| Accuracy | 3±1% | 2.9±0.6% | 15±2% | 29±3% |
| Mean $L_2$ dist. | 3.36±0.03 | 3.43±0.05 | 2.9±0.1 | 2.4±0.2 |
| Model | – | S64 | S128 | S256 |
| Accuracy | – | **71±2%** | **57±5%** | **46±3%** |
| Mean $L_2$ dist. | – | 1.9±0.3 | 3.0±0.4 | 3.5±0.1 |

In Figure 2, it can be observed that binary and full-precision models perform somewhat similarly for the first few iterations of the CWL2 attack, but beyond 10–20 iterations, the accuracy of full-precision models drops off quickly, regardless of having performed adversarial training. We note that PGD, defined with respect to the $L_\infty$ norm, makes no claim of increasing robustness to $L_2$ attacks, such as CWL2. Interestingly, it can be seen that the binary model benefited from adversarial training considerably when evaluated at 10 to 100 attack iterations, while the full-precision model did not. These benefits eventually disappear to within the margin of random error after continuing to 1000 iterations, as recommended by Carlini & Wagner (2017). At this point, both B and B+ had accuracy of 19±3%, by which time the full-precision models had long flatlined at zero. Meanwhile,

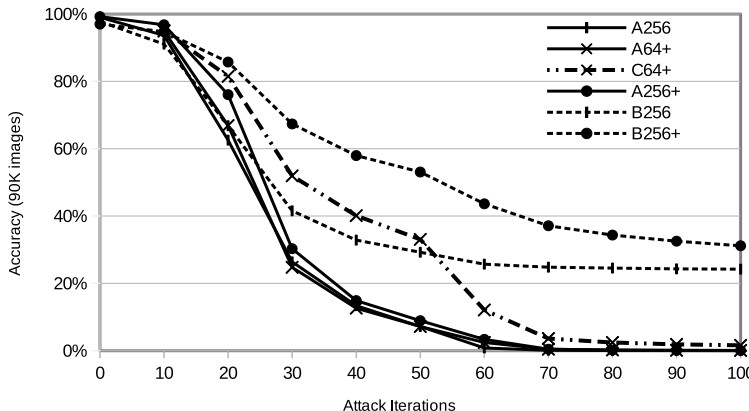

Figure 2: Accuracy of full-precision (A), binary (B), and scaled binary (C), models subject to targeted Carlini-Wagner $L_2$ attacks of increasing strength on MNIST dataset. Models A/B256+ and A/C64+ were trained with 20 and 40 iterations of PGD, respectively.

S64 maintained $38 \pm 3\%$ accuracy after 1000 iterations, nearly double that of the deterministically quantized models. Running these attacks to 1000 iterations was two orders of magnitude more time consuming than training these models from scratch (without PGD training); therefore we believe this targeted attack represents a fairly substantial level of effort on behalf of the adversary.

## 3.2 BLACK-BOX ATTACKS

We run the substitute model training procedure from Papernot et al. (2017b) using CleverHans v2.0.0, for both MNIST and CIFAR-10 datasets with and without FGSM adversarial training. As a substitute model, we use a two-layer MLP with 200 hidden units and ReLU activations. The substitute is trained on 150 images withheld from the test set, and augmented by perturbing the images in the direction of maximal variability of the substitute model, as defined by the Jacobian. Six epochs of data augmentation with $\lambda = 0.1$ were used in combination with 10 substitute model training epochs after each augmentation step. The oracle was again trained for 15 epochs for MNIST, and 20 epochs for CIFAR-10.

Table 4: Accuracy of Oracle models on adversarial MNIST examples transferred from a Papernot et al. (2017b) style smooth substitute model black-box misclassification attack. Images from test set attacked with FGSM with $\epsilon = 0.3$. FGSM adversarial training indicated by '+' suffix, and 20 iterations of PGD training for 40 epochs by '+*' suffix.

| Filters | 64 | 128 | 256 |
|---------|-----|------|------|
| A | 79±1% | 78±4% | 73±5% |
| A+ | 73±2% | 76±4% | 80±2% |
| A+* | **95.8±0.4%** | **96.4±0.3%** | **96.7±0.3%** |
| B | 46±5% | 55±4% | 39±3% |
| B+ | 42±2% | 52±3% | 50±6% |
| C | 51±4% | 56±6% | 54±10% |
| C+ | 65±9% | 72±6% | 70±2% |
| C+* | **94.7±0.3%** | **95.6±0.1%** | **96.4±0.4%** |
| S+* | 56±1% | 68±2% | 77.9±0.7% |

Results for the black-box experiment on the MNIST dataset are shown in Table 4. Full-precision networks had a moderate advantage over undefended binary models B and C. Only the highest capacity full-precision model benefited from FGSM adversarial training, while the scaled binary model benefited regardless of capacity. There was a small positive relationship between accuracy and capacity for both A and C when trained with PGD, and there was almost no loss in accuracy in this setting

after binarization. PGD was more effective than stochasticity here as it leads to learning a more optimal decision boundary, rather than confusing an adversary with dynamic gradient information.

Table 5: Accuracy of Oracle models on clean and adversarial CIFAR-10 examples transferred from a Papernot et al. (2017b) style smooth substitute model black-box misclassification attack with FGSM and $\epsilon = 0.3$. FGSM adversarial training indicated by '+' suffix.

| Filters | Accuracy Type | 64 | 128 | 256 |
|---------|---------------|-----|------|------|
| A | Clean | 64.4±0.6% | 64.2±0.3% | 63.2±0.9% |
| | Transfer | 23±2% | 22±1% | 22±1% |
| A+ | Clean | 64.1±0.6% | 64±1% | 65.2±0.4% |
| | Transfer | 16.8±0.7% | 22±1% | 19±1% |
| C | Clean | 62±1% | 64±1% | 61±1% |
| | Transfer | 20.2±0.6% | 20±1% | 21±1% |
| C+ | Clean | 57.3±0.2% | 61±1% | 63±2% |
| | Transfer | **24.1±0.5%** | **25.8±0.5%** | **27.6±0.9%** |

## 4  DISCUSSION

We suspect that plain BNNs implement two different kinds of gradient masking. We discovered the first by tracking the L1 norm of the hidden layer activations and unscaled logits. BNNs operate with larger range and variance than 'normal' networks, which can be explained by virtue of convolving inputs with greater magnitude ($\pm 1$) compared with the typically small values taken by weights and activations. For our 64 kernel CNN, the logits were about $4\times$ larger than the scaled or full-precision networks. This is analogous to the more complex defensive distillation procedure in which the model to be secured is trained with soft-labels generated by a *teacher* model. When training the teacher, a softmax temperature, $T \gg 1$ is used. The distilled model is trained on the labels assigned by the teacher and using the same $T$. At test time, the model is deployed with $T = 1$, which causes the logits to explode with respect to their learned values. The logits saturate the softmax function and cause gradients to vanish, leading FGSM and JSMA to fail at a higher rate. However, this defense is defeated with a close enough guess for $T$, or via a black box attack (Carlini & Wagner, 2017).

The second type of gradient masking is less easily overcome, and has to do with gradients being inherently discontinuous and non-smooth, as seen in Figure 3 of Appendix B. We believe that this effect is what gives scaled BNNs an advantage over full-precision with respect to targeted attacks. Even more importantly, through a regularization effect, the decision boundary for the MLP with binary units (Figure 3) better represents the actual function to be learned, and is less susceptible to adversarial examples. But why does gradient masking have a disproportionate effect when attacking compared with training on clean inputs? Models 'A' and 'B' were trained to within 1.2% test accuracy, while 'B' had improvements of 9.0% and 29.5% on JSMA and CWL2 attacks respectively, corresponding to $8\times$ and $25\times$ difference in accuracy, respectively, for adversarial vs. clean inputs. For JSMA, the performance gap can be attributed to the sub-optimality of the attack as it uses logits rather than softmax probabilities. Furthermore, to achieve its L0 goal, pairs of individual pixels are manipulated which is noisy process in a binarized model.

The success of model 'S' with stochastically quantized weights in its third convolutional layer against iterative attacks is more easily explained. Adversarial examples are not random noise, and do not occur in random directions. In fact, neural networks are extremely robust to large amounts of benign noise. An iterative attack that attempts to fool our stochastically quantized model faces a unique model at every step, with unique gradients. Thus, the direction that minimizes the probability of the true class in the first iteration is unlikely to be the same in the second. An iterative attack making $n$ steps is essentially attacking an ensemble of $n$ models. By making a series of small random steps, the adversary is sent on the equivalent of a wild goose chase and has a difficult time making progress in any particularly relevant direction to cause an adversarial example.

## 5   CONCLUSION

We have shown that for binarized neural networks, difficulty in training leads to difficulty when attacking. Although we did not observe a substantial improvement in robustness to *single step* attacks through binarization, by introducing stochasticity we have reduced the impact of the strongest attacks. Stochastic quantization is clearly far more computationally and memory efficient than a traditional ensemble of neural networks, and could be run entirely on a micro-controller with a pseudo random number generator. Our adversarial accuracy on MNIST against the best white-box attack (CWL2) is $71\pm2\%$ (S64+) compared with the best full-precision model $1.8\pm0.9\%$ (A256+). Black-box results were competitive between binary and full-precision on MNIST, and binary models were slightly more robust for CIFAR-10, which we attribute to their improved regularization. Beyond their favourable speed and resource usage, we have demonstrated another benefit of deploying binary neural networks in industrial settings. Future work will consider other types of low-precision models as well as other adversarial attack methods.

ACKNOWLEDGMENTS

The authors wish to acknowledge the financial support of NSERC, CFI and CIFAR. The authors also acknowledge hardware support from NVIDIA and Compute Canada. We thank Brittany Reiche for helpful edits and suggestions that improved the clarity of our manuscript.

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

# A    CLEAN TEST ERROR RATES

| $K_{Layer1}$ / Model | 64 | 128 | 256 |
|---|---|---|---|
| A | 1.2±0.2% | 1.1±0.1% | 1.06±0.2% |
| A+ | 0.99±0.02% | 1.0±0.1% | 1.03±0.03% |
| B | 2.3±0.1% | 2.2±0.2% | 2.3±0.2% |
| B+ | 2.0±0.2% | 1.73±0.09% | 1.9±0.1% |
| C | 1.3±0.2% | 1.2±0.1% | 1.2±0.1% |
| C+ | 1.3±0.1% | 1.21±0.09% | 1.08±0.05% |

Table 6: Error on clean MNIST test set for models with varying capacity and precision. A is full-precision, B is binary, and C is binary with a learned scalar applied to the ReLU in hidden layers. All models were trained with Adam for 15 epochs with a batch size of 128 and a learning rate of 1e-3. For adversarially trained models, we used 20 iterations of PGD (Madry et al., 2017) with $\epsilon = 0.3$ for the last 5 epochs.

# B    MLP TOY AND PROBLEM

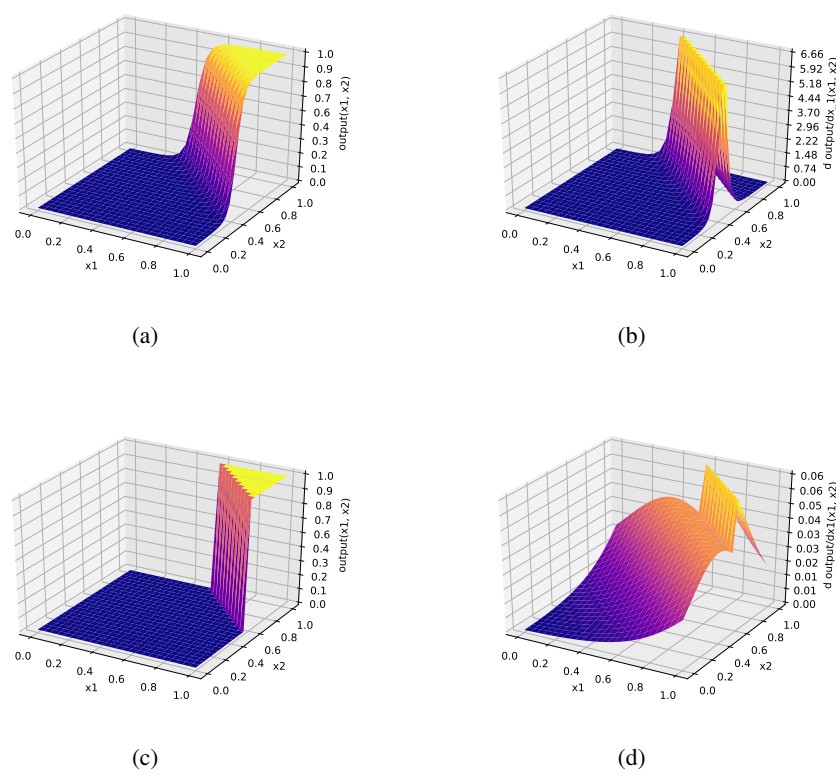

(a)

(b)

(c)

(d)

Figure 3: Decision surface for a three layer MLP with two hidden units in first two layers, and sigmoid output neuron [(a) and (c)]. Corresponding forward derivative with respect to input $x_2$ [(b) and (d)]. Full-precision model [(a) and (b)] and model with a binarized hidden layer [(c) and (d)].

We reproduce the toy problem in Papernot et al. (2015) of learning the two-input logical AND function with a simple MLP having two neurons in each layer. The only difference between our experiment and the original is that we train a 3-hidden-layer MLP (as opposed to 2-layers) with the Adam optimizer for 1k epochs, with a learning rate of 0.1. We use 3 layers since this is the smallest

number of layers where the middle one can be quantized without directly touching the input or output, which would adversely impact learning. Here, a "quantized" layer means that its weights and activations are thresholded to +1 and -1, and a straight through estimator (Bengio et al., 2013) is used to backpropagate gradients for learning.

All configurations in the AND experiment learn a reasonable decision boundary; however, the MLPs with a single quantized hidden layer had highly non-linear forward gradients, as can be seen in Figure 3(d). As training progresses, the forward derivative was highly dynamic and took on a variety of different shapes with sharp edges and peaks. When the MLP was allowed more capacity by doubling the number of hidden units (see Figure 4), the forward derivative was almost entirely destroyed. If one was to use this information to construct a saliency map, only two regions would be proposed (with poor directional information), and once exhausted there would be no further choices more insightful than random guessing.

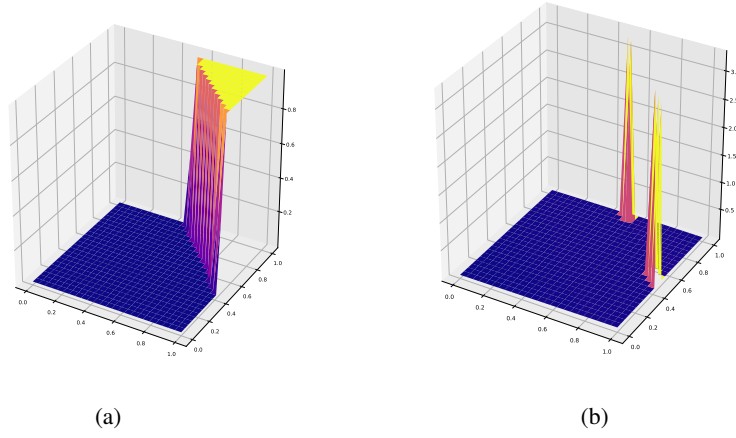

(a)                                    (b)

Figure 4: (a) Decision surface for a 3 layer MLP with four hidden units in first two layers, one output neuron, and quantized middle layer. (b) Corresponding forward derivative.

## C  VISUALIZING LOGITS WITH SCALING FACTORS

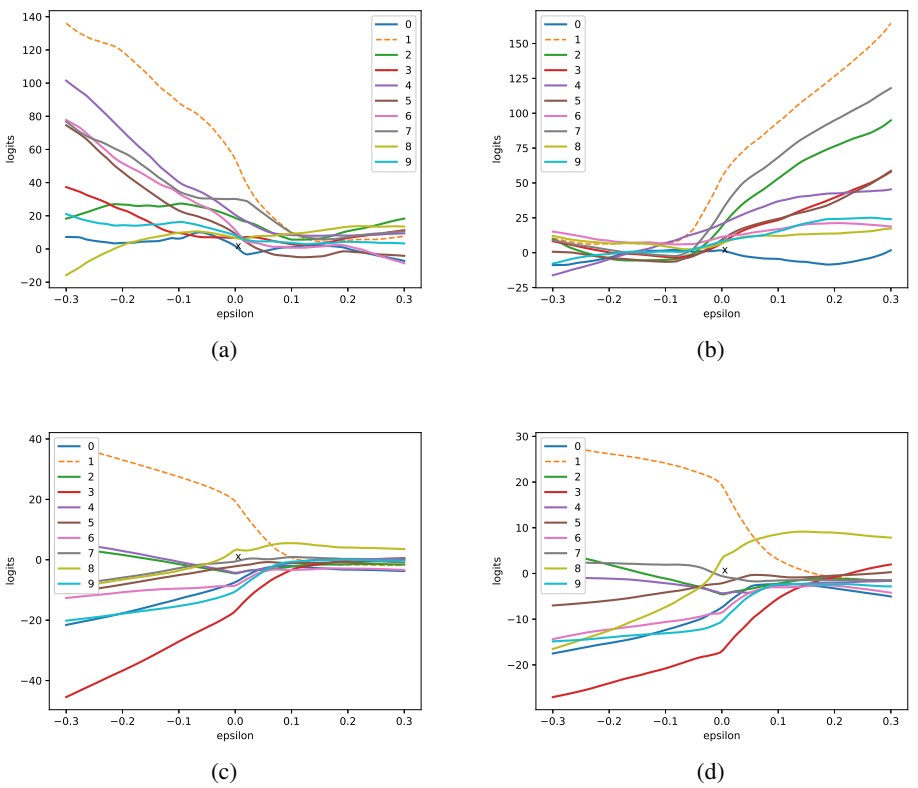

(a)  (b)

(c)  (d)

Figure 5: We reproduce the plot from (Goodfellow et al., 2015) by evaluating the logits of non-scaled binary [(a) and (b)] and full-precision [(c) and (d)] neural networks for an MNIST digit with varying degrees of FGSM perturbation. Note that the true class of the digit is "1" in this instance. The softmax temperature, $T$, was 0.6, 0.7, 0.1, and 1.0 in each of (a), (b), (c), and (d) respectively.

In Figure 5 we compare the logits of full-precision and binary networks under varying degrees of FGSM perturbation. We noticed that for softmax temperature $T$ between 0.6–0.7 the direction in which increasing the perturbation causes an adversarial example flips. We observe no similar effect for full-precision models. Additionally the full-precision logits respond to scaling in an approximately linear manner, whereas there is very little change in logits for the binary case apart from the 180 degree flip. We used values of $\epsilon$ in the range of actual attacks conducted in the paper, however the piecewise linear effect from (Goodfellow et al., 2015) is still there for $\epsilon$ with large absolute value.

