# OpenReview forum: "Attacking Binarized Neural Networks"
_ICLR.cc/2018/Conference — Accept (Poster)_

### Official Review · AnonReviewer1 · 2017-11-26

**Rating:** 7
**Confidence:** 3

**Review:**


This paper starts by gently going over the concept of adversarial attacks on neural networks (black box vs white box, reactive vs proactive, transfer of attacks, linearity hypothesis), as well as low-precision nets and their deployement advantages.
Adversarial examples are introduced as a norm-measurable deviation from natural inputs to a system. We are reminded of adversarial training, and of the fact that binarized nets are highly non linear due to the nature of their weights and activations.

This paper then proposes to examine the robustness of binarized neural networks to adversarial attacks on MNIST and CIFAR-10.

The quantization scheme used here is v32 Conv2D -> ReLU -> BNorm -> sign -> bit Conv2D -> ReLU -> Scalar -> BNorm -> sign, but sign is really done with the stochastic quantization method of Courbariaux et al, even at test time (in order to make it more robust).

What the experimental results show:
- High capacity BNNs are usually more robust to white-box attacks than normal networks, probably because the gradient information that an adversary would use becomes very poor as training progresses.
- BNNs are harder to properly train with adversarial examples because of the polarized weight distribution that they induce
- Against black-box attacks, it seems there is little difference between NNs and BNNs.

Some comments and questions:
- In figure 1 I'm not sure what "Scalar" refers to, and it is not explained in the paper (nor could I find it explained in Papernot et al 2017a).
- Do you adopt the "Shift based Batch Normalizing Transform" of Courbariaux et al? If not, why?
- It might be worth at least _quickly_ explaining what the 'Carlini-Wagner L2 from CleverHans' is rather than simply offering a citation with no explanation. Idem for 'smooth substitute model black-box misclassificiation attack'. We often assume our readers know most of what we know, but I find this is often not the case and can discourage the many newcomers of our field.
- "Running these attacks to 1000 iterations [...], therefore we believe this targeted attack represents a fairly substantial level of effort on behalf of the adversary." while true for us researchers, computational difficulty will not be a criterion to stop for state actor or multinational tech companies, unless it can be proven that e.g. the number of iterations needs to grow exponentially (or in some other unreasonable way) in order to get reliable attacks.
- "MLP with binary units 3 better", 'as in Fig.' is missing before '3', or something of the sort.
- You say "We postpone a formal explanation of this outlier for the discussion." but really you're explaining it in the next paragraph (unless there's also another explanation I'm missing). Training BNNs with adversarial examples is hard.
- You compare stochastic BNNs with deterministic NNs, but not with stochastic NNs. What do you think would happen? Some of arguments that you make in favour of BNNs could also maybe be applied to stochastic NNs.

My opinions on this paper:
- Novelty: it certainly seems to be the first time someone has tackled BNNs and adversarial examples
- Relevance: BNNs can be a huge deal when deploying applications, it makes sense to study their vulnerabilities
- Ease of understanding: To me the paper was mostly easy to understand, yet, considering there is no page limit in this conference, I would have buffed up the appendix, e.g. to include more details about the attacks used and how various hyperparameters affect things.
- Clarity: I feel like some details are lacking that would hinder reproducing and extending the work presented here. Mostly, it isn't always clear why the chosen prodecures and hyperparameters were chosen (wrt the model being a BNN)
- Method: I'm concerned by the use of MNIST to study such a problem. MNIST is almost linearly separable, has few examples, and given the current computational landscape, much better alternatives are available (SVHN for example if you wish to stay in the digits domain). Concerning black-box attacks, it seems that BNNs less beneficial in a way; trying more types of attacks and/or delving a bit deeping into that would have been nice. The CIFAR-10 results are barely discussed.

Overall I think this paper is interesting and relevant to ICLR. It could have stronger results both in terms of the datasets used and the variety of attacks tested, as well as some more details concerning how to perform adversarial training with BNNs (or why that's not a good idea).

---

### Official Review · AnonReviewer3 · 2017-11-27

**Rating:** 7
**Confidence:** 4

**Review:**

1) Summary
This paper proposes a study on the robustness of one low-precision neural networks class - binarized neural networks (BNN) - against adversarial attacks. Specifically, the authors show that these low precision networks are not just efficient in terms of memory consumption and forward computation, but also more immune to adversarial attacks than their high-precision counterparts. In experiments, they show the advantage of BNNs by conducting experiments based on black-box and white-box adversarial attacks without the need to artificially mask gradients.


2) Pros:
+ Introduced, studied, and supported the novel idea that BNNs are robust to adversarial attacks.
+ Showed that BNNs are robust to the Fast Gradient Sign Method (FGSM) and Carlini-Wagner attacks in white-box adversarial attacks by presenting evidence that BNNs either outperform or perform similar to the high-precision baseline against the attacks.
+ Insightful analysis and discussion of the advantages of using BNNs against adversarial attacks.

3) Cons:
Missing full-precision model trained with PGD in section 3.2:
The authors mention that the full-precision model would also likely improve with PGD training, but do not have the numbers. It would be useful to have such numbers to make a better evaluation of the BNN performance in the black-box attack setting.


Additional comments:
Can the authors provide additional analysis on why BNNs perform worse than full-precision networks against black-box adversarial attacks? This could be insightful information that this paper could provide if possible.


4) Conclusion:
Overall, this paper proposes great insightful information about BNNs that shows the additional benefit of using them besides less memory consumption and efficient computation. This paper shows that the used architecture for BBNs makes them less susceptible to known white-box adversarial attack techniques.

---

### Official Review · AnonReviewer2 · 2017-12-08
**elegant method, less convincing experiments**

**Rating:** 6
**Confidence:** 5

**Review:**

his work presents an empirical study demonstrating that binarized networks are more robust to adversarial examples. The authors follow the stochastic binarization procedure proposed by Courbariaux et al. The robustness is tested with various attacks such as the fast gradient sign method and the projected gradient method on MNIST and CIFAR.

The experimental results validate the main claims of the paper on some datasets. While reducing the precision can intuitively improve the robustness, It remains unclear if this method would work on higher dimensional inputs such as Imagenet. Indeed:

(1) state of the art architectures on Imagenet such as Residual networks are known to be very fragile to precision reduction. Therefore, reducing the precision can also reduce the robustness as it is positively correlated with accuracy.

(2) Compressing reduces the size of the hypothesis space explored. Therefore, larger models may be needed to make this method work for higher dimensional inputs.

The paper is well written overall and the main idea is simple and elegant. I am less convinced by the experiments.

---

### Public Comment · (anonymous) · 2017-12-16
**Replication of 'Attacking Binarized Neural Networks'**

1) Introduction: As a ﬁnal project for the course ”Applied Machine Learning” at McGill University, Canada, we were tasked with reproducing a paper submitted to the International Conference on Learning Representation (iclr.cc). We chose the paper ”Attacking Binarized Neural Networks” because of its application to mass-produced and available technology. Low-precision networks can be deployed on more cost-effective hardware and can therefore be more widely utilized.

2) Analysis of Paper: The paper is well written and provides many novel approaches and ﬁndings:
• The authors of this paper conducted many experiments.
Each experiment was well deigned and focused on one aspect of the model.
• The paper demonstrated that binary neural networks are often more robust against adversarial attacks.
• The paper found that binary neural network were harder to properly train with adversarial examples than the full-precision network.

However, there were limitations to this paper:
• In the paper, the author stated that they ran the training and attacking in both MNIST and CIFAR-10 datasets, however, in the white-box attack portion, they only showed the results of performing MNIST dataset. They did not mention the CIFAR-10 dataset.
• The authors applied the low-precision neural network to the datasets only in a relatively low dimension. It will be more convincing if they can test this neural network architecture on some higher dimensional datasets, such as Imagenet.

3) Reproduction Methodology: The reproduction is realized with TensorFlow and CleverHans. We conducted attacks on similarly modeled BNNs and full-precision networks and compared their performance. Both white-box and black-box attacks were reproduced running under similar parameters in the original work.
In addition, we further veriﬁed BNNs’ robustness against adversarial attacks by experimenting on CIFAR-10 dataset. Details of our reproduction can be seen in paper linked below.

4) Reproduction Results: To replicate the original paper’s white-box attack methods we ran both Fast-Gradient Sign Method (FGSM) and Carlini-Wagner L2 attacks on different neural network setups. From these attacks, we found that we were able to replicate the original papers ﬁndings. Most of our results consistently fell within the ranges that the author has listed. However, for some tests, we have received accuracies drastically different than those reported in the original paper. These inconsistencies may be due to us using slightly different model parameters as they were not well speciﬁed in the original paper. They may also be reduced by rerunning our replicated tests until we have more conﬁdence in our values.
Replication of Black-Box attacks consisted of running the same substitute model training procedure from Papernot et al. using CleverHans v2.0.0 on the MNIST dataset with FGSM adversarial training.
Similar to our White-box replication, some of our Black-box results had accuracies similar to those originally reported. However, when we attacked the binary neural network with learned scalar, our result differed far from the authors’ experimental results.
Full-precision networks had a moderate advantage over binary model classes B and C, which was similar to the authors’ result. However, our binary neural network with learned scalar performed worse than binary network without the learned scalar. We think that the difference is due to due to the different number of adversarial instances.

5) Conclusion: We reproduced the main ﬁndings of the paper Attacking Binarized Neural Networks. We found out that neural networks with low-precision weights and activations, such as binarized neural networks, would indeed improve robustness against some adversarial attacks like FGSM and Carlini-Wagner L2.

6) Nota Bene: A copy of the original paper can be found at https://goo.gl/rQvXig.

---

### Author Response · Authors · 2018-01-03
**Response to all reviewers (part 2)**

R1 - Computational effort of running CWL2 to 1000 iterations

The purpose of our statement was to provide some context for the level of effort spent attacking a network in terms of the effort required to train it originally. We agree that this is not a formal notion of security, although for high dimensional problems such as ImageNet, an attack that is an order of magnitude more expensive than training from scratch could be prohibitive for the layperson.

R1 - More detail on attacks

We agree that more detail regarding the attacks makes the paper more accessible. As such, we have added a brief description of the Carlini-Wagner L2 attack. We note that the Papernot “smooth substitute model-black box attack” was briefly outlined in Section 3.2 and do not feel that we have much more space to elaborate given the target paper length.

R3 - Full-precision model with PGD training in Section 3.2

We can confirm that the full-precision model (A) does indeed perform much better with PGD adversarial training rather than with FGSM, but so does the scaled binarized model (C), which we did not report originally in Table 4. We find no significant difference between models A and C when PGD training is used for models of varying capacity. Please refer to the latest revision for the updated Table 4.

R3 - Can the authors provide additional analysis on why BNNs perform worse than full-precision networks against black-box adversarial attacks? This could be insightful information that this paper could provide if possible.

We wish to clarify that BNNs are not worse than NNs against black-box attacks, which should be more clear in the updated Table 4 (see C+*), and from Table 5 (C+). When using delayed PGD training, and scaling binarized activations by a single small tunable parameter per layer, similar performance to full-precision with PGD training is achieved on MNIST. For CIFAR-10, the scaled BNN with FGSM adversarial training (C+) achieved 8.6% higher black-box accuracy than full-precision with FGSM training (A+) for the high capacity model, and C+ maintained a small edge over A+ at the lowest capacity tested. A possible explanation for the improvement of the scaled binarized model on CIFAR-10 is that it has limited representation power to cheat by learning image statistics and other non-salient consistencies between training and test sets, an explanation inspired by Jo & Bengio, 2017. We have observed this effect in preliminary MNIST experiments with simple classifiers and a small set of feature-preserving frequency domain transformations applied to either training or test split, but a more detailed explanation is still in progress.

R1 - In figure 1 I'm not sure what "Scalar" refers to, and it is not explained in the paper (nor could I find it explained in Papernot et al 2017a).

We have updated the second paragraph in Section 3 with additional detail about the nature and purpose of this scalar. The original description referred to a “scaling factor” but we have updated the language. This scalar does not come from Papernot et al., rather it is a modification to vanilla BNNs that reduces the range of hidden activations so they align more closely to NNs. This prevents numerical instabilities at the softmax layer, which has implications for gradient based attacks, and leads to improved accuracy on clean inputs as originally reported by Tang et al., 2017.

R1 - Do you adopt the "Shift based Batch Normalizing Transform" of Courbariaux et al? If not, why?

We did not use shift based batch normalization (SBN) as this was viewed as a performance trick for reducing the number of multiplications required by “vanilla” batch normalization (BN) during training. The original Courbariaux paper reported no loss in accuracy when using SBN rather than BN for the same datasets. Traditional aspects of BNN performance were outside the scope of this paper.

Additional References:

Thomas Tanay and Lewis Griffin. A Boundary Tilting Perspective on the Phenomenon of Adversarial Examples, 2016 -- https://arxiv.org/abs/1608.07690

Yichuan Tang and Ruslan Salakhutdinov. Learning Stochastic Feedforward Neural Networks, 2013  -- http://www.cs.toronto.edu/~tang/papers/sfnn.pdf

Tapani Raiko, Mathias Berglund, Guillaume Alain, and Laurent Dinh. Techniques for Learning Binary Stochastic Feedforward Neural Networks, 2014  -- https://arxiv.org/abs/1406.2989

Jason Jo and Yoshua Bengio. Measuring the tendency of CNNs to learn surface statistical regularities, 2017 -- https://arxiv.org/abs/1711.11561

---

### Author Response · Authors · 2018-01-03
**Response to all reviewers (part 1)**

We thank the reviewers for their positive and constructive feedback. We believe that we have addressed all of the main questions and concerns in the most recent revision of the paper. These are detailed below:

R2 - Higher dimensional data

To confirm that our findings hold for higher dimensional data, and further show that performance on clean inputs does not necessarily translate to the same on adversarial examples, we conducted ImageNet experiments with AlexNet and ResNet-18 using 1-4 bits for weights and activations. Models were trained as in DoReFa-Net by Zhou et al., 2016, with the addition of L2 weight decay on the first layer (conv0) of all models which isn’t quantized. This L2 norm penalty is inspired by the boundary tilting perspective (Tanay & Griffin, 2016). A regularization constant of 1e-5 was used for AlexNet and 1e-4 for ResNets.

We found that:

For AlexNet, despite some accuracy degradation for clean inputs, all low-precision variants had the same or less top-1 error against FGSM, and all but two had less top-5 error across a typical range of perturbation magnitudes (epsilon in [2.0 - 16.0]). A binarized AlexNet had 4.5% and 8.0% less top-1 and top-5 error respectively than a full-precision equivalent for epsilon=2.0, and performed the same or better when sweeping across the full range of epsilon.

ResNet experienced a 6.5% and 4.2% reduction in top-1 and top-5 error respectively on clean inputs when going to 2-bits, however this performance gap shrinks to within +/- 0.2% for FGSM with epsilon=4.0 (in favour of low-precision for top-1 error, and in favour of full-precision for top-5). A binarized ResNet was slightly less optimal than the 2-bit case, but still managed to reduce the performance gap on clean inputs, resulting in 3.4% higher top-5 error, but 0.4% lower top-1 error, for FGSM with epsilon=4.0, as compared to full-precision.

The small 3x3 kernels in ResNets are less likely to be unique for very low bitwidths, as there are only 512 possible binary 3x3 kernels, vs 65k possible binary 4x4 kernels. This could explain some of the differences between binarizing AlexNet which uses a mix of large and small kernels, vs ResNet which makes exclusive use of small kernels.

Although one of the main contributions of DoReFa-Net was to train with low bitwidth gradients, we’re reporting the 32-bit gradient case here as this is what was done originally in our paper, and is more representative of “black-box” vulnerability in an ideal FGSM transfer attack. Models trained with low bitwidth gradients (e.g 6-bits) further reduced top-5 error under FGSM by 6-7% on AlexNet, however this gain was found to be caused by gradient masking, as it was overcome by subsequently attacking with 32-bit gradients.

Our preference is to report these results here informally to address the questions/concerns of R2, to keep the paper at a reasonable length, and because the use of mixed precision and larger dataset could be seen as a departure from the original scope of the paper (c.f. the instructions to authors). The majority of related works report CIFAR-10 and MNIST since adversarial robustness is generally an unsolved problem even in low-dimensions (e.g see https://github.com/MadryLab/mnist_challenge and https://github.com/MadryLab/cifar10_challenge). Several of the targeted attacks and adversarial training methods used in the paper (e.g PGD, CWL2, Papernot black-box transfer attack) are currently very slow to run on ImageNet.

In the time since submission, we have also been analyzing the difference between compression by low-precision parameters and compression by other means such as pruning. Pruning introduces sparsity which can result in loss of rank in weight matrices, effectively removing coordinate axis with which to express the decision boundary and causing it to collapse and lie near the data, leaving the model vulnerable to adversarial examples. Low-precision is less likely to result in loss of rank unless used in conjunction with very small kernels.

R1 - Stochastic BNN vs Stochastic NN

We conducted but did not report some additional experiments for the full-precision case where weights are sampled from a Gaussian distribution with a learned mean. This type of model achieved 20% accuracy against 100 iterations of CWL2, compared to 70% for SBNNs. Having weights flip signs is more destructive and noisy wrt the progress of an iterative attack. Weights sampled from a Gaussian are less likely to change sign between iterations. In the literature outside low-precision implementations, typically stochastic NNs refers to stochastic activations (e.g. [Tang and Salakhutdinov 2013, Tapani et al. 2014]) and this is likely what R1 was referencing. While we agree with the reviewer that networks with stochastic activations are indeed a relevant comparison, we have not yet carried out the experiments.

---

### Decision · Program_Chairs · 2018-01-29
**ICLR 2018 Conference Acceptance Decision**

**Decision:**

Accept (Poster)

**Comment:**

Paper was well written and rebuttal was well thought out and convincing.

The reviewers agree that the paper showed BNNs were good (relatively speaking) at resisting adversarial examples. Some question was raised about whether the methods would work on larger datasets and models. The authors offered some experiments in this regard in the rebuttal to this end. Also, a public comment appeared to follow up on CIFAR and report correlated results.